# Synthesis and Assessment of Novel Sustainable Antioxidants with Different Polymer Systems

**DOI:** 10.3390/polym16030413

**Published:** 2024-02-01

**Authors:** Agathe Mouren, Eric Pollet, Luc Avérous

**Affiliations:** BioTeam/ICPEES ECPM, UMR CNRS 7515, Université de Strasbourg, 25 rue Becquerel, 67087 Strasbourg CEDEX 2, France; agathe.mouren@etu.unistra.fr (A.M.); eric.pollet@unistra.fr (E.P.)

**Keywords:** antioxidants, TPU, polyolefin, sustainable additive, phenolic acids

## Abstract

Antioxidants are essential to the polymer industry. The addition of antioxidants delays oxidation and material degradation during their processing and usage. Sustainable phenolic acids such as 4-hydroxybenzoic acid or 3,4-dihydroxybenzoic acid were selected. They were chemically modified by esterification to obtain various durable molecules, which were tested and then compared to resveratrol, a biobased antioxidant, and Irganox 1076, a well-known and very efficient fossil-based antioxidant. Different sensitive matrices were used, such as a thermoplastic polyolefin (a blend of PP and PE) and a purposely synthesized thermoplastic polyurethane. Several formulations were then produced, with the different antioxidants in varying amounts. The potential of these different systems was analyzed using various techniques and processes. In addition to antioxidant efficiency, other parameters were also evaluated, such as the evolution of the sample color. Finally, an accelerated aging protocol was set up to evaluate variations in polymer properties and estimate the evolution of the potential of different antioxidants tested over time and with aging. In conclusion, these environmentally friendly antioxidants make it possible to obtain high-performance materials with an efficiency comparable to that of the conventional ones, with variations according to the type of matrix considered.

## 1. Introduction

According to the International Union of Pure and Applied Chemistry (IUPAC), an additive is defined as “A substance added to a polymer, usually a minor compound in the mixture formed that modifies the properties of the polymer” [1]. Except for certain additives such as plasticizers and fillers/reinforcements, the additives are chemical compounds added in relatively low concentrations into polymer matrices. Without the addition of the different additives, polymers would have virtually no properties for usage and application. For instance, plastics would exhibit very poor resistance over time. The market for additives is therefore very large. By 2022, it has been estimated at over USD 55 billion [2]. Additives can be divided into different categories [3]. This is a sector of innovation and numerous literature reviews have already been published reporting some recent developments in this field [4,5,6].

Among the different additives, antioxidants play a crucial role in the formulation of plastic materials. The addition of antioxidants delays oxidation and therefore the degradation of materials during their processing (oxidation under thermomechanical treatment) and during their use [5]. Antioxidants are classified into two categories according to their modes of action with (i) primary antioxidants, which have the ability to neutralize all or some of the radicals (R∙, RO∙, ROO∙, HO∙) by proton transfer, and (ii) secondary antioxidants, which break down the formed hydroperoxides (ROOH).

Thermoplastic polyolefins (TPOs), such as polypropylene (PP) or polyethylene (PE), and thermoplastic polyurethane (TPU) are among the world’s most widely used thermoplastics. Together, they account for more than half of all thermoplastics produced. They stand out for their excellent performance, with a wide range of mechanical and thermal properties, good chemical resistance, and easy processing. However, if not formulated, these neat polymers could not be so widely used alone. This is because the long carbon chains of TPO or the long polyol part of TPU are sensitive to thermo-oxidative degradation, particularly when subjected to thermomechanical treatment with oxygen, light or heat, mainly during processing, but also during their lifetime [7]. The oxidative degradation of polymers is initiated by the formation of free radicals on the polymer chain due to the breaking of carbon bonds. These free radicals then react with the oxygen present in the air, initiating an autocatalyzed reaction that will ultimately generate chain breakdowns, resulting in a loss of polymer properties [8]. 

For decades, the use of primary antioxidants to stabilize TPOs has proved to be an effective strategy for preserving the macromolecular structure and molar mass of these polymers and extending their lifetimes [9]. These molecules act by scavenging the free radicals formed during oxidation reactions, thus interrupting the cascade of chain degradation reactions. Among these, the hindered phenolic antioxidants attract particular attention due to their chemical structures and mechanisms of action. The phenolic group possesses a labile hydrogen which is essential in the proton transfer mechanism. When the phenolic antioxidant encounters a free radical, the hydroxyl (OH) group yields its labile hydrogen to form a stabilized phenolic radical, thanks to the delocalization potential of the aromatic ring which stabilizes the molecule by resonance [10].

In the field of antioxidants, the term “irganox” has become a generic one. A large range of Irganox^©^ products is produced by BASF (Ludwigshafen, Germany). Within this very broad offer, Irganox 1076 (octadecyl 3 (3,5 di tert-butyl 4 hydroxyphenyl) propanoate) is the best-known antioxidant used in industry, thanks to the presence of sterically hindered phenolic groups and its long alkyl group. This specific structure makes it a versatile antioxidant and thermal stabilizer, for a wide range of polymer systems. Irganox 1076 effectively scavenges free radicals resulting from polymer oxidation, thus interrupting chain degradation reactions. Its chemical structure does not only effectively neutralize these radicals, but also prevents their formation thanks to its ability to inhibit thermal and photo-oxidative degradation. Furthermore, the presence of a long alkyl group enhances its solubility and compatibility with several polymer matrices. This molecule is stable at high temperatures, with a high molar mass (531 g mol^−1^), enabling it to retain its effectiveness during polymer melt processing. This rather high molar mass provides low volatility associated with a good compatibility with different polymers [11]. Nevertheless, studies have shown that this type of antioxidant can migrate into drinking water or food [12,13,14]. Then, these molecules can be accumulated into the human body, affecting health. A study describes Irganox 1076 as potentially toxic to reproduction [15]. Different studies have also shown that certain phenolic antioxidants widely used in industry, such as butylated hydroxyanisole (BHA) or butylated hydroxytoluene (BHT), are potentially carcinogenic [16,17].

Although phenols are widely present in compounds of natural origin (lignin, carotenoids, flavonoids, etc.), most antioxidants are fossil-based. However, some antioxidants of sustainable origin are more effective than many of the fossil-based bulky phenols used in industry. But most bio-based antioxidants, such as lignins, cause an undesirable coloration of the polymeric materials [18]. Moreover, their solubility is limited, especially in apolar matrices such as TPO, due to their intrinsic polarity [19].

Resveratrol, a polyphenol found in various plant sources, is an aromatic antioxidant known to the pharmaceutical and food industries, which is attracting growing interest in the field of polymers with reported antioxidant activities superior to BHT [20]. It can also be obtained by microbial engineering from various sustainable carbon substrates (biomass, plastic waste, …) such as glucose or ethanol [21,22].

Phenolic acids such as 4-hydroxybenzoic acid (4-HB) or 3,4-dihydroxybenzoic acid (3,4-DHB) can be obtained from biomass or by a chem-biotech approach from various substrates, including plastic waste [23,24]. They have been known to have antioxidant activities [25]. Nevertheless, molecules bearing a phenol in the meta position to the acid or ester group, appear to have weaker antioxidant activity than those in the ortho or para positions [26]. Also, the introduction of electron-donating or electron-accepting groups at different positions on the aromatic ring, close to the phenol groups, can enhance the molecule’s antioxidant potential [27]. These are often o-methoxy or o-hydroxyl groups and alkyl esters [28]. Furthermore, it has been reported in the literature that the presence of aliphatic OH groups can increase the antioxidant capacity of a molecule [29]. In addition, the presence of primary OH groups linked to an aromatic ring bearing a phenol, as in the case of tyrosol or hydrotyrosol, also increases the antioxidant capacity of the molecule [30]. This is also the case with lignins and their antioxidant properties [18]. It has also been reported that the number of phenolic OH groups, as well as the length of the alkyl chain linking the aromatic ring to the primary OH group, impact the antioxidant performance [30].

A variety of diols or triols can be chemically and/or biologically obtained from various sustainable resources, such as biomass or waste plastics. For example, ethylene glycol (EG) is a component of poly(ethylene terephthalate) (PET) that can be obtained directly by chemical or enzymatic depolymerization, as can BDO from poly(butylene succinate) (PBS) or poly(butylene adipate terephthalate) (PBAT) [31,32]. These molecules as well as glycerol or 1,6-hexanediol (HDO) can also be obtained by fermentation from engineered microorganisms using different carbonaceous substrates from biomass, such as sugars [33,34,35]. All these molecules can also be chemically obtained starting from different biomasses. 

In the context of this study, different esterification reactions with polyols such as EG, BDO, HDO or glycerol were performed on phenolic acid compounds to provide a primary OH group and an ester group, potentially increasing their overall antioxidant efficiency. This modification strategy could also improve (i) the thermal stability by increasing the molar masses of 4-HB and 3,4-DHB, (ii) the solubility in polymer matrices, and (iii) the matrix affinity by reducing their loss by exudation/migration or volatilization. The initial antioxidant activities of the starting molecules and synthesized products are then compared with those of resveratrol and Irganox 1076 (as reference). Figure 1 presents the chemical structures of these molecules. As antioxidant–polymer matrix interactions can impact antioxidant activity, TPO and TPU matrix formulations were prepared with different antioxidants at varying concentrations, and then characterized. An accelerated aging protocol was set up to evaluate variations in polymer properties and estimate the evolution of the potential of different antioxidants over time and with aging.

## 2. Materials and Methods

### 2.1. Materials and Chemicals

EG (99.8%), HDO (97%), BDO (99%), and 2,2-diphenyl 1-picrylhydrazyle (DPPH) were obtained from Sigma Aldrich (Saint-Quentin-Fallavier, France). Glycerol (≥99%). Para-toluene sulfonic acid (PTSA) (monohydrate, 99%), methanol (MeOH) (analytical reagent grade, ≥99%), THF (analytical reagent grade, ≥99%), 4-hydroxybenzoic acid (4-HB) (99%), 1,4-benzenedimethanol (BDM) (99%) and 3,4-dihydroxybenzoic acid (3,4-DHB) (97%) were purchased from Thermo Scientific (Illkirch, France). Deuterated dimethylsulfoxide (DMSO-d_6_) (99.8%) was bought from Eurisotop (Saint Aubin, France). Ethyl acetate (99.9%) was obtained from VWR International (Rosny-sous-bois, France. Resveratrol was purchased from Fluorochem (Hadfield, UK). Methylene diphenyl 4,4′-diisocyanate (MDI) and Suprasec 1306 were obtained from Huntsman (Krakow, Poland). Poly(tetrahydrofuran) (PTHF) with weight average molar mass of M_w_ = 2000 g mol^−1^ (PTHF 2000) was obtained from BASF (Ludwigshafen, Germany) under the PolyTHF^©^ tradename. TPO Hifax CA 10A was obtained from LyondellBasell (Rotterdam, The Netherlands). It is a propylene and ethylene copolymer (M_n_ = 42 kg mol^−1^, Đ = 10.4), conventionally used in industrial applications such as building and automotive industries. THANOX 1076 (octadecyl 3-(3,5-di-tert-butyl-4-hydroxyphenyl) propanoate), referred to in the following study under the generic name «Irganox 1076», was obtained from Green Chemicals (Desio, Italy). 

### 2.2. Antioxidants Synthesis

The chemical structures of the antioxidants and their synthesis routes are shown in Figure 1 and Figure 2, respectively.

For the synthesis of 4-hydroxybutyl 3,4-dihydroxybenzoate (BDO-3,4DHB), 5 g of 3,4-DHB was introduced into a two-neck round bottom flask and dissolved in 10 molar equivalents of BDO, used as a reactive solvent. Five percent PTSA was added to the solution, then stirred and heated to 100 °C under vacuum, for 24 h. The evolution of the reaction was monitored by ^1^H NMR. After cooling to room temperature, the product was precipitated in cold water and recovered by filtration as a white crystalline solid. 

The synthesis of 2-hydroxyethyl 4-hydroxybenzoate (EG-4HB) was carried out in the same way, with 4-hydroxybenzoic acid (4-HB) and EG as the acid and the diol, respectively. The 4-hydroxybutyl 4-hydroxybenzoate (BDO-4HB), 6-hydroxyhexyl 4-hydroxybenzoate (HDO-4HB) and 2,3-dihydroxypropyl 4-hydroxybenzoate (glycerol-4HB) were also prepared with the same pathway, with BDO or HDO as diol, respectively, and glycerol as the triol.

### 2.3. Formulations Based on TPO and TPU Matrices

Different formulations based on a TPO (a PE and PP blend) and a TPU were prepared with different antioxidants at various contents. The different formulations are given in Table 1. TPO formulations were prepared by thermo-mechanical route in molten state. Formulations of TPU with antioxidants were prepared in solvent media and dried by solvent casting.

To prepare the formulations based on the TPO matrix, the chamber of an internal mixer (Haake PolyLab OS) was heated to 170 °C, i.e., around 20 °C above the melting temperature (T_m_) of the TPO matrix. A measure of 40 g of TPO was introduced into the chamber and mixed, with a rotor speed of 50 rpm, for a few minutes, until torque stabilization. The appropriate amount of antioxidant was then introduced, and the rotor speed increased to 70 rpm, for 10 min, before stopping the heating and recovering the mixture. Films with a thickness of 1 mm were then produced using a hot press (LabTech Engineering Company Ltd., Welkenraedt, Belgium) and 10 × 10 cm^2^ molds. For that, around 13 g of polymer was first preheated for 3 min, then compressed between the plates at 170 °C and 160 bar for 8 min, and finally cooled down for 5 min at room temperature.

For the TPU-based formulation, a conventional 2-step synthesis was carried out. A prepolymer of MDI and PTHF 2000 is synthesized in the first step at an NCO/OH (I_NCO_) value of 2 for one hour at 70 °C under mechanical stirring at 200 rpm in a three-neck round bottom flask. The synthesis route is shown in Figure 3. As a chain extender, BDM is added in solution to give a final TPU with I_NCO_ = 1. The mixture is stirred for 5 min at 200 rpm and then poured into a square Teflon mold. TPU was then placed in a vacuum oven at 50 °C overnight to finalize the reaction. Then, 500 mg TPU was dissolved in 5 mL THF at 60 °C and stirred until completely dissolved. A measure of 0.3 wt% of antioxidant was added into the solution, which was again stirred for 5 min, then the mixture was poured into a glass Petri dish. The solvent was then evaporated overnight in a fume hood. The final material was placed in a vacuum oven at 50 °C for 6 h to remove the last traces of solvent.

### 2.4. Characterization Techniques and Procedures

^1^H NMR spectra were obtained with a Bruker 400 MHz spectrometer in DMSO-d_6_. Sixteen scans were collected at 25 °C. Calibration was performed using the chemical shift of DMSO-d_6_ (δH = 2.50 ppm).

The antioxidant activity of the synthesized molecules was quantified by DPPH assay adapted from the literature [28]. The DPPH reagent was dissolved in methanol to give a concentration of 0.09 mM. Antioxidant solutions were prepared at different concentrations in methanol. The radical scavenging activity of the antioxidant was determined by mixing 30 µL of antioxidant solution with 3 mL of DPPH solution in a quartz vessel. The solution was agitated on a vortex mixer and placed for 24 h in the dark, to ensure complete reaction. Sample absorbance was measured at 517 nm using a Shimadzu UV 2600 spectrometer. A blank was run in a quartz cell containing methanol. The percentage of DPPH trapping activity was calculated using Equation (1).
(1)$${\%}_{inhibition\;}=\frac{{A_{blank}-A}_{sample}}{A_{blank}}\times 100$$

$A_{blank}$ and $A_{sample}\;$ are the absorbances of the blank and the sample, respectively. The antioxidant concentration required for 50% inhibition of the DPPH free radical (=EC50) was calculated by linear regression of the curve plotting % DPPH inhibition vs. concentrations.

Uniaxial tensile tests were performed on an Instron 5567H (Elancourt, France) with a 10 kN cell. Experiments were carried out at 20 °C with a constant crosshead speed of 20 mm min^−1^. Dumbbell-shaped samples with dimensions of approx. 30 × 5 × 1 mm^3^ were produced for each system. Young’s modulus (E), strength at break (σ_max_) and elongation at break (ε_max_) were determined as average values from 5 specimens.

The thermal stability of the synthesized molecules was assessed by thermogravimetric analysis (TGA) using a Hi Res TGA Q5000 from TA Instruments (Guyancourt, France), to evaluate their thermal stability during the plastic processing steps. Samples were heated in dry air to 180 °C at a rate of 30 °C min^−1^, then this temperature was maintained for 20 min.

The polymers and the formulations were also studied by TGA. Measurements were carried out in dry air atmosphere using a Hi Res TGA Q5000 from TA Instruments. Measures of 1 to 3 mg of neat polymer or formulation were heated to 650 °C at 20 °C min^−1^. Temperatures corresponding to 5% mass losses are indicated as T_5%_. Temperatures corresponding to peak maxima in mass loss derivatives (DTG) are indicated as T_deg_. 

Differential scanning calorimetry (DSC) was performed using a Discovery DSC 25 from TA Instruments (Guyancourt, France). Approximately 10 mg of polymers were placed in sealed aluminum capsules and analyzed under nitrogen flow (50 mL min^−1^). A three-step procedure with a ramp of 10 °C min^−1^ was applied as follows: (i) heating from 90 to 200 °C to erase the thermal history, (ii) cooling to 90 °C (iii) heating from 90 to 200 °C. The glass transition and melting temperatures (T_g_ and T_m_, respectively) were determined during the second heating step. 

The antioxidant activity of the different formulations was determined by the oxidation induction time (OIT) test determined by DSC. Approximately 2 mg of formulation was placed in an open aluminum pan, as described in the literature [36]. For TPO formulations, the sample was heated to 200 °C at a rate of 10 °C min^−1^ under nitrogen, followed by an isotherm of 5 min. Nitrogen was then replaced by dry air for an isotherm at 200 °C for 300 min. For TPU formulations, after preliminary tests, the final temperature was set at 190 °C and the first isotherm (under nitrogen) lasted 5 min, until the heat flow was stabilized, after which nitrogen was replaced by air for 30 min. The OIT corresponds to the time elapsed between exposure to the oxidizing gas and the start of degradation represented by the appearance of an exothermic peak on the heat flow curves.

The yellow index (YI) of TPO formulations after aging was determined using a colorimeter (Konica Minolta CR 200 Chroma Meter) with CIELAB (*L**, *a**, *b**) color space. The *L** and *b** values were used to determine the yellowing of the material through the value of the yellow index (YI) determined with Equation (2) [37].
(2)$$\;{YI}_{\;}=142.86\frac{b^{*}}{L^{*}}$$

The aging of TPO films with antioxidants was tested by placing the samples in an oven at 85 °C for 85 days, under continuous air flow according to the norm EN 14575:2005 [38]. After each aging period, the samples were removed from the oven and cooled down to room temperature. The samples are then analyzed using various techniques to assess and monitor changes in their properties (OIT, YI, tensile tests).

## 3. Results

### 3.1. Antioxidants’ Synthesis and Characterizations

Hydroxy-esters syntheses from 4-HB and 3,4-DHB were carried out by esterification with diols, such as EG, BDO and HDO, or a triol such as glycerol, to potentially increase polymer matrix compatibility, thermal stability, and antioxidant efficiency. Esterification and carbon chain addition is an increasingly common approach to the functionalization of phenolic antioxidants, as is the addition of primary aliphatic hydroxyl groups [30,39]. Proton NMR spectra of the corresponding products are available in Appendix A.

In contrast to syntheses previously described in the literature [40,41], the products were obtained in reactive solvent with a “mild” catalyst and a purification step based on precipitation in water, in line with green chemistry principles [42]. However, this purification step reduces the yields obtained in this study (by around 40%), compared with the literature, which employs mostly toxic apolar solvents, such as hexane, for liquid-phase extractions. 

### 3.2. Evaluation of Antioxidant Activity

The antioxidant activity of the synthesized molecules, resveratrol and Irganox 1076, was quantified using the DPPH assay. This is a common method for determining the antioxidant activity of molecules as radical scavengers [43]. This test is based on the ability of antioxidants to donate hydrogen atoms to the stable free radical DPPH (purple), leading to its conversion to a non-radical form (yellow). The reduction in DPPH absorbance, at 517 nm, indicates the molecule’s radical scavenging capacity. Figure 4A illustrates DPPH consumption as a function of the antioxidant structure and the relative contents. EC50 values were obtained by linear regression on these curves, and are presented in Figure 4B.

The steeper the drop in the curves, the more efficient the antioxidant is. In consequence, Irganox 1076 and resveratrol have the best DPPH consumption rates. Resveratrol possesses an unsaturation which stabilizes the deprotonated molecule by resonance. In addition, the 4-hydroxystilbene group has been shown to stabilize the free radical and promote a variety of resonance structures, thus enhancing its antioxidant power [44]. Irganox 1076 has a highly sterically hindered phenolic OH, which is less easily accessible, as well as a linear fatty alkyl chain. This reduces its ability to form intermolecular hydrogen bonds, making it more effective as an antioxidant additive [45].

Antioxidant activities were also analyzed for 3,4-DHB and BDO-3,4DHB. According to the literature, the presence of two OH groups in the ortho and para positions to the acid or the ester group, as for 3,4 DHB and its derivatives, would favor their antioxidant activities [46]. Contrary to some expectations, the esterification with BDO slightly reduced the antioxidant power of 3,4-DHB. On the other hand, none of the molecules synthesized from 4-HB exhibited antioxidant activity in the concentration range studied, irrespective of the length of the grafted alkyl chain or the number of aliphatic OH groups per molecule. The low antioxidant activity of 4-HB and its derivatives has already been reported in the literature [39,47]. According to these authors, this is due to the single, poorly hindered phenolic OH group.

### 3.3. Thermal Stability of the Formulations

The thermal stability and volatility of the various molecules were studied by isothermal TGA. This enabled us to validate or not the development of formulations prepared in an internal mixer at 170 °C, without degradation and/or major loss of volatiles. The overall thermal treatment with a temperature set at 180 °C for 20 min, resulted in very severe treatment conditions. Results are shown in Figure 5.

The results show that 4-HB and 3,4-DHB are not totally stable or non-volatile at 180 °C with a mass loss of 55 and 18%, respectively, after 20 min. These results are due to the low molar masses of the compounds and then their high volatilities. BDO-4HB and HDO-4HB also lack thermal stability, with mass losses of 16 and 25%, respectively, unlike EG-4HB. This could be explained by the difference in chain length between the ester and the aliphatic OH. Indeed, a longer chain (like BDO or HDO, and unlike EG) brings better flexibility, which could lead to more rapid molecular rearrangements or configuration changes at high temperatures. The chemical structure of the molecules thus impacts their stability over time at high temperatures [48]. In addition, glycerol-4HB also shows good thermal stability, which seems to be linked to the presence of three OH groups (a phenolic and primary and secondary aliphatic ones) that could form hydrogen bonds stabilizing the structure. The association of BDO onto 3,4-DHB to form BDO-3,4DHB significantly enhances the molecule’s thermal stability. 

Irganox 1076, resveratrol and BDO-3,4DHB displayed good thermal stability at 180 °C under air, with a good potential to scavenge the free radicals that can be generated under heat treatment, as shown by the DPPH test.

## 4. Analysis of the Matrices and the Formulations

### 4.1. TPO Case

As TPOs are particularly sensitive to oxidation phenomena, they are the most stabilized polymers with antioxidants. It should be noted that the antioxidant efficiency of a molecule can vary depending on the polymer matrix due to potential “antioxidant–polymer” interactions or affinities [49]. To evaluate this point, formulations with a TPO matrix were carried out, with different types of antioxidants at varying concentrations (Table 1). Only the antioxidants BDO-3,4 DHB, resveratrol and Irganox 1076 were evaluated, according to the previously obtained results from the thermal stability at 180 °C and DPPH tests. The formulation characteristics are also shown in Table 2.

The thermal degradation of TPO-based formulations in air was assessed by TGA. Indeed, thermo-oxidative degradation of polyolefins creates free radicals that cause polymer chain scissions. This degradation thus leads to volatile compounds resulting in a rapid mass loss in TGA curves. The mass loss curves for formulations with 3 wt% antioxidants are shown in Figure 6. T_5%_ and T_deg_ values are reported in Table 2. The results show a positive impact of antioxidants on the thermal stability under air of the TPO, with only small variations between the different antioxidants tested. This behavior is in line with results previously reported in the literature on the addition of lignin-based antioxidants on polyolefins [50]. The significant temperature shift (ca. 103 °C) of the main mass loss observed for the different formulations therefore indicates the efficiency of the antioxidants to capture free radicals delaying the thermo-oxidative degradation.

The antioxidant activity of these materials was particularly characterized by the OIT test. The results are shown in Figure 7. The aim of this test is to observe the impact of the addition of an antioxidant on the polymer oxidation. The role of the antioxidant is to delay the oxidation of the material in which it is incorporated. Thus, the higher the OIT, the more effective the antioxidant is.

For Irganox 1076 and resveratrol, a very significant increase in OIT was observed with increasing antioxidant concentration, confirming their good antioxidant activities. Resveratrol showed a similar antioxidant activity to Irganox 1076, due to its unsaturation and OH group in para position to the ester, which offers a wide variety of resonance forms, increasing its antioxidant efficiency, as observed in the DPPH assay. These results are in line with the literature [51,52].

BDO-3,4DHB also showed a significant increase in OIT with increasing antioxidant contents, which also confirms its antioxidant effect, in agreement with the DPPH results. This is mainly due to the two OH groups on the aromatic ring, which increase free radical scavenging potential and the antioxidant efficiency. 

The OIT of TPO + 1.5% BDO-3,4DHB was comparable to those of TPO + 0.3% Irganox 1076 and TPO + 0.3% Resveratrol. Thus, an antioxidant activity comparable to that of commercial antioxidants can be achieved with five times more BDO-3,4DHB. According to the literature, this is likely due to the phenolic groups in the meta position to the ester group of BDO-3,4DHB, which may reduce the molecule’s antioxidant activity [53]. The latter results are in line with those of the DPPH tests, indicating that the additives are well suited to the used matrix. 

Uniaxial tensile tests and DSC analyses were carried out to determine the impact of antioxidant incorporation in a TPO matrix and on its thermal and mechanical properties. DSC and tensile results are shown in Appendix A, and Figure 8, respectively. 

Small variations are observed in T_g_ and T_m_, depending on the type of antioxidant and their respective concentrations. This shows the low impact of antioxidant additions on the overall thermal properties of TPO. The same applies to mechanical properties in uniaxial tensile tests.

According to the literature, one of the main disadvantages of using biobased and/or sustainable antioxidants is the coloration they impart to the material. Indeed, these antioxidants are generally brown or even yellow in color, and therefore generate unwanted coloring. The yellow index (YI) of the formulations was therefore determined by colorimetry and values are displayed in Figure 9. For TPO formulations based on the Irganox 1076, the YI is lower than that of the neat matrix, indicating a slight whitening effect of the additive. However, for resveratrol and BDO-3,4DHB, the YI of the formulations increases significantly with antioxidant concentration. The YI of formulations with 0.5% of resveratrol and BDO-3,4DHB were, respectively, five and seven times higher than the neat matrix.

### 4.2. TPU Case

The literature shows that the stability of TPU is mainly linked to the long polyol stability [54]. The main thermo-oxidative degradation mechanisms are similar to those of TPO [55]. Thus, the mechanisms for restricting these phenomena are equivalent, including the use of primary antioxidants as radical scavengers. Then, for the TPU, only the best antioxidants (selected from the previous study with TPO) will be studied, i.e., Irganox 1076 and resveratrol.

In this case, TPU has been synthesized in a classic two-step protocol and obtained from different potentially sustainable building blocks (PTHF, MDI and BDM). A long polyether polyol PTHF (2000 g mol^−1^) was used. PTHF can be synthesized from THF, itself produced from BDO which can be derived from sustainable resources such as biomass or by chemical and/or enzymatic depolymerization at end-of-life of polymers such as PU, PBS or PBAT [32,56]. BDM can be obtained by hydrogenolysis of PET or poly(butylene terephthalate) (PBT) [32]. MDI can be produced from aniline derived from sugarcane [57]. The thermal properties (TGA and DSC) of the synthesized TPU are given in Table 2.

From TPU, formulations containing 0.3 wt% of Irganox 1076 or resveratrol were produced by mixing in solvent, followed by an evaporation step to obtain the TPU-based materials. Figure 10 shows the mass loss curves and corresponding derivative curves (DTG) obtained by TGA analysis of TPU and its formulations with 0.3 wt% of antioxidant. T_5%_ and T_deg_ values are given in Table 3. Three conventional stages of TPU degradation are visible for the different analyzed materials. They are represented by three DTG peaks [58]. The first stage of degradation around 330 °C corresponds to the decomposition of TPU into diisocyanates and polyols. It is less important for systems containing antioxidants. The second stage is attributed to the degradation of polyols. This stage is around 380 °C for neat TPU, and 420 °C for TPU with 0.3 wt% antioxidants. This represents a gain of 40 °C attributed to the addition of antioxidant. The third and final stage (around 560 °C) corresponds to residue degradation, and it is similar for all materials. 

According to the literature, the TPU soft segments (SS), such as the PTHF part, are the most impacted by oxidative degradation [59]. Oxidation begins with the formation of a radical in the α position of the ether bond, which induces the formation of hydroperoxides. The latter decomposes with chain scission and the generation of a free radical which can continue the degradation [60]. This autoxidation mechanism, particularly visible during the second stage of degradation attributed to the degradation of SS, is slowed by the presence of Irganox 1076 and resveratrol, which capture the free radicals formed. In addition, the MDI segment of the TPU which is a part of the hard segment (HS) can also be affected by thermo-oxidative degradation. According to the literature, it can be converted to benzophenone following reactions involving free radicals [61]. The presence of resveratrol or Irganox 1076 reduces this phenomenon by scavenging the free radicals. Then, different gains in degradation temperatures can be obtained for the formulations with antioxidant compared to the neat matrix.

The OIT of the neat TPU and the formulations based on 0.3% of Irganox 1076 or 0.3% of resveratrol, were determined by DSC, in air at 190 °C. The curves obtained are shown in Figure 11. The OIT of the neat matrix was 3 min, which is in line with the literature values for this type of polymer [62]. An improvement in the OIT of up to 6 min was obtained with the addition of 0.3% of Irganox 1076. Moreover, no significant exotherm was observed for the formulation containing 0.3% of resveratrol. These results also demonstrate the excellent antioxidant activity of these molecules in a TPU matrix, with the thermo-oxidative stability provided by resveratrol far superior to the values obtained in the literature for stabilized TPUs [63].

### 4.3. Analysis of an Accelerated Aging Test on TPO-Based Formulations

Since TPOs are the most sensitive polymers to oxidation phenomena, particularly thermal oxidation, they were chosen for a specific accelerated aging study. Monitoring the aging of TPO formulations enables us to study variations in the material properties and antioxidant activity with the TPO matrix over time, and to follow the influence of the antioxidant type and contents on these properties. Accelerated aging involved keeping the formulations in air at 85 °C for 85 days and carrying out periodic analyses. The OITs of the various systems were measured after different aging times as shown in Figure 12.

After 85 days, all formulations containing an antioxidant still showed higher OITs than the neat matrix. This confirms that the antioxidant was still active for all the additive types and studied contents. Irganox 1076 appeared to be more effective than resveratrol at contents above 0.5%. Resveratrol seemed most effective at low concentrations. However, BDO-3,4DHB showed stable results from 31 days onwards, whereas for the other systems, most OITs declined sharply between the end of the first month and the third one. Then, despite its lower initial efficacy, BDO-3,4DHB could hold promise for the long-term stabilization of TPO against oxidation, in line with certain applications.

The YI was also measured during aging for the various TPO formulations. A slight yellowing of the neat TPO was noted during aging, linked to an intrinsic degradation phenomenon of the polymers. As shown in Figure 13A, formulations containing Irganox 1076 had YIs below 15, virtually identical to the matrix alone, for all antioxidant contents. Irganox therefore does not intrinsically generate yellowing. In this case, the stabilized TPO did not show any significant change in color over time. The materials seemed to yellow slightly during the first two months of aging, and then appeared to stabilize. 

The YI of materials based on resveratrol (Figure 13B) or BDO-3,4DHB (Figure 13C) increased sharply over time, with values reaching 45. In the case of resveratrol, this is due to the presence of two aromatic rings per molecule and a styrene double bond. The concentration of conjugated bonds in these molecules is therefore higher than in Irganox 1076. According to the literature, these conjugated bonds are responsible for the coloration of TPOs, as well as the presence of two OH groups per aromatic ring, as in BDO-3,4DHB. Indeed, after radical capture, these antioxidants are stabilized in a quinone form that is highly chromophoric [64]. Moreover, the synthesis of BDO-3,4DHB requires a temperature of 100 °C for 24 h, which induces an initial notable coloration for this molecule. 

All these results show the great interest of Irganox 1076 and explain its high consumption in industry. In addition to the addition of colorants or pigments, the coloration induced by these sustainable primary antioxidants could also be reduced by the addition of phosphorus compounds (secondary antioxidants), which break down hydroperoxides and thus reduce quinone formation [64]. In addition, it could also be possible to valorize the color change in the materials as an asset, for example as a color indicator of the aging of formulated materials [65].

The mechanical properties of the formulated materials were also analyzed during the aging. The results are reported in Appendix A. It has been reported in the literature that autoxidation phenomena in TPO induce chain scissions which ultimately strongly degrade the mechanical properties of the polymers [66]. Nevertheless, in our present study, the variations in mechanical properties over time are not representative. Furthermore, there is no significant difference between formulations with and without additives under the conditions of the norm EN 14575:2005 [38]. These standardized conditions do not therefore appear to be sufficiently discriminating for these systems.

## 5. Conclusions

This study focused on the valorization of phenolic acid-based molecules potentially derived from sustainable resources, such as biomass or plastic waste. Chemical modifications were performed on 4-HB and 3,4-DHB, using molecules obtained from sustainable resources, with the aim to improve properties such as thermal stability and antioxidant capacity. To this end, esterification with polyols from sustainable resources were successfully carried out. The efficiency of these synthesized molecules was compared with that of resveratrol, also obtained from sustainable resources, and Irganox 1076, a hindered phenol widely used in industry obtained from fossil resources. Overall, BDO-3,4DHB, EG-4HB, glycerol-4HB, resveratrol and Irganox 1076 showed good thermal stability. 

DPPH tests quantifying their ability to scavenge free radicals were also carried out. 3,4-DHB, BDO-3,4DHB, Irganox 1076 and resveratrol showed very good antioxidant efficiency, with the best results obtained with resveratrol. This is due to the presence of the unsaturation and 4-hydroxystilbene group, which ensures delocalization and stabilization of the formed radical. Similarly, 3,4-DHB and its derivative carry two closely related phenolic groups in the ortho and para positions to the acid or ester groups, which enhance the antioxidant potential. The 4-HB and its derivatives, on the other hand, did not show radical scavenging capacity, due to their single, weakly hindered phenolic group in the para position to the acid or ester groups. 

In the case of TPO and TPU, the additives improved thermal stability by scavenging the free radicals responsible for the oxidation of polyolefin, or polyether chains (in the case of TPU). OITs have also been greatly improved, even at low concentrations. Resveratrol showed better results than Irganox 1076, which is the most widely used in the industry. However, a marked coloration of the matrix is observed with the biobased additives, while Irganox 1076 shows a slight whitening effect. 

Formulations based on TPO, a matrix known to be particularly sensitive to these phenomena, were subjected to accelerated aging tests, in air at 85 °C for 85 days, to determine the effectiveness of antioxidants over time and under thermal stress. The OIT was measured throughout the aging process. Resveratrol showed very good long-term stability, particularly at low content. BDO-3,4DHB was shown to be promising for long-term antioxidant activity. Nevertheless, the yellowing of formulations with the bio-based additives increased sharply during accelerated aging, unlike those with Irganox 1076. The yellowing is due to the presence of the two OH groups per aromatic ring, which are stabilized as quinones by reaction with hydroxyperoxides. However, depending on the intended application, this yellowing is not always detrimental and could be greatly limited by the addition of optical brighteners or dyes and pigments.

In addition to studies on migration, (eco)toxicity, etc., different systems based on synergies with secondary antioxidants could be investigated. However, according to the literature, this often involves toxic phosphites. In addition, as it is done industrially, the analysis of combined short- and long-term efficiency systems could be carried out by mixing different antioxidants. For example, BDO-3,4DHB, playing the role of long-term stabilizer could be associated with another short-lasting stabilizer but displaying a higher initial efficiency.

This study represents an important step in the search for solutions to improve the stability and durability of industrial polymeric materials by exploiting the antioxidant properties of aromatic molecules derived from renewable resources. By enhancing thermal stability and reducing the effects of oxidation, these molecules could help extend the useful life of polymers, offering significant environmental benefits in the very broad field of long-term plastic applications. 

## Figures and Tables

**Figure 1 polymers-16-00413-f001:**
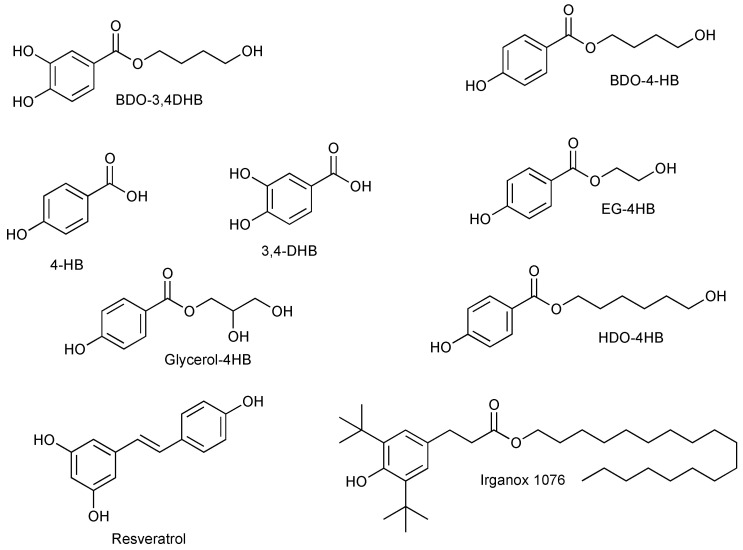
Chemical structure of the different potential antioxidants.

**Figure 2 polymers-16-00413-f002:**
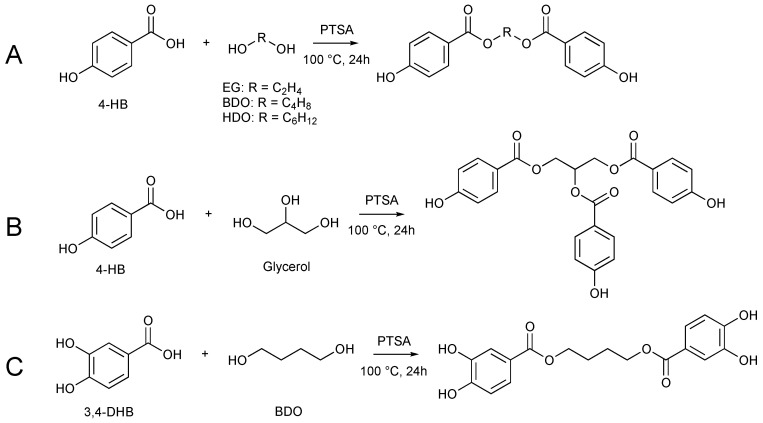
Synthesis pathways for (**A**) EG-4-HB, BDO-4-HB, HDO-4-HB, (**B**) glycerol-4-HB and (**C**) BDO-3,4-DHB.

**Figure 3 polymers-16-00413-f003:**
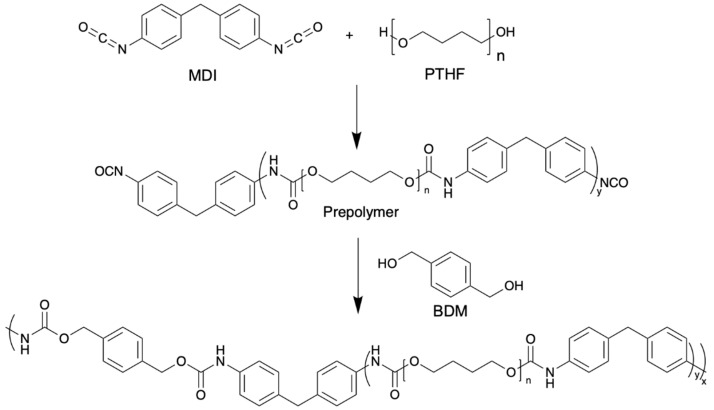
TPU synthesis based on a two-step protocol.

**Figure 4 polymers-16-00413-f004:**
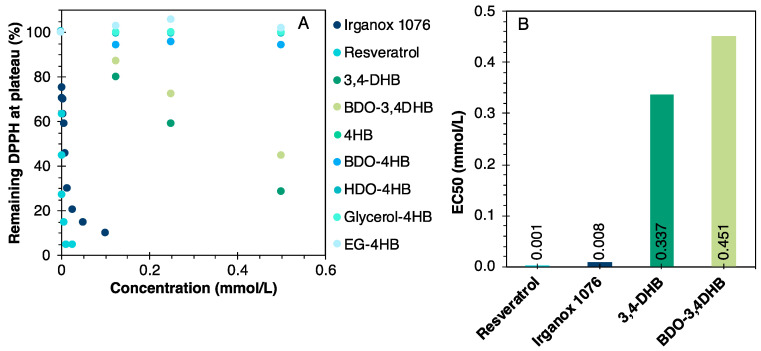
Results of the DPPH tests, determined with the different additives. (**A**) DPPH consumption as a function of the antioxidant structure and the relative contents. (**B**) EC50 values were obtained by linear regression on these curves.

**Figure 5 polymers-16-00413-f005:**
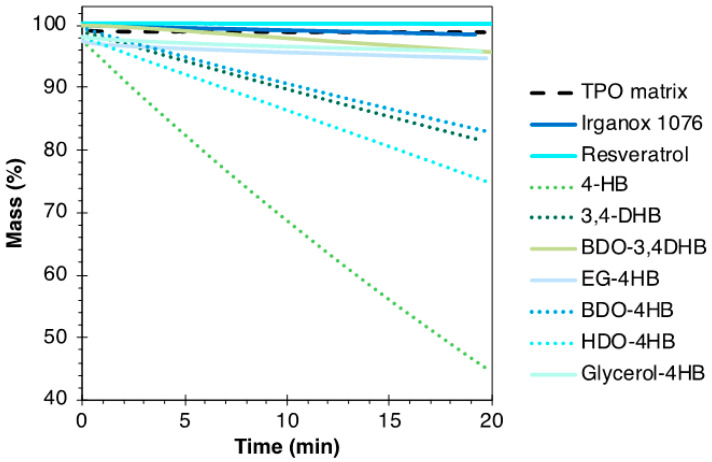
TGA results at 180 °C in air for the TPO matrix and the various studied molecules.

**Figure 6 polymers-16-00413-f006:**
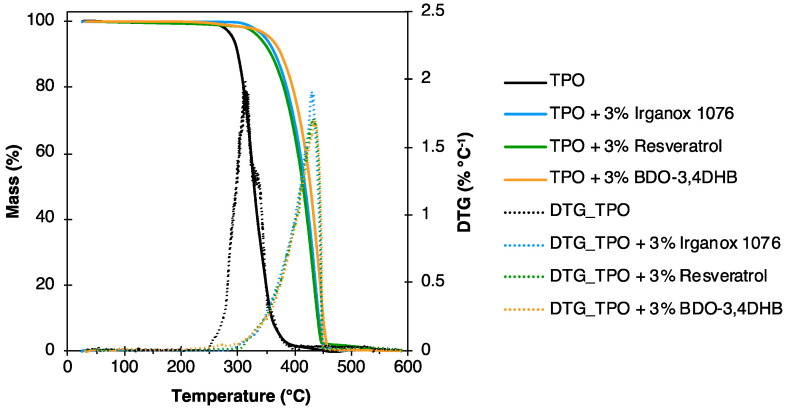
Results of TGA and DTG of TPO-based formulations with 3 wt% antioxidants.

**Figure 7 polymers-16-00413-f007:**
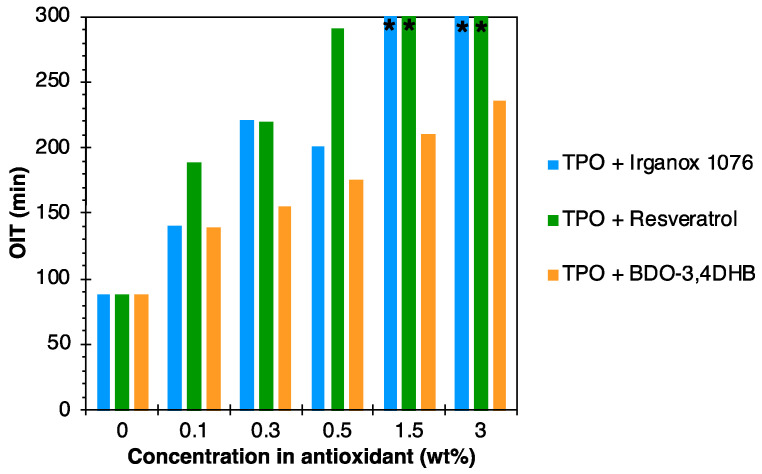
OIT results for TPO-based formulations. (*) corresponds to OITs higher than 300 min.

**Figure 8 polymers-16-00413-f008:**
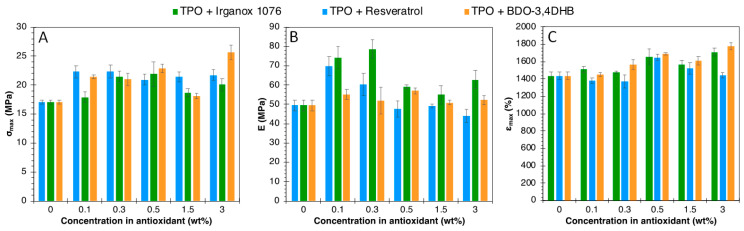
Uniaxial tensile test results on TPO formulations. (**A**) maximum strength, (**B**) Young’s modulus and (**C**) Elongation at break values.

**Figure 9 polymers-16-00413-f009:**
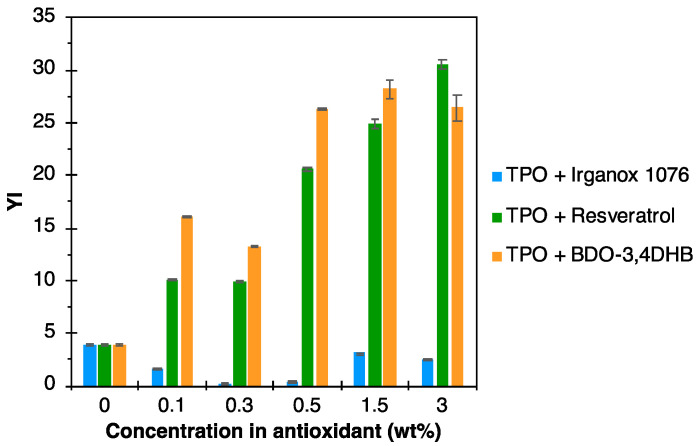
YI values of TPO-based formulations as a function of the contents of different types of antioxidants.

**Figure 10 polymers-16-00413-f010:**
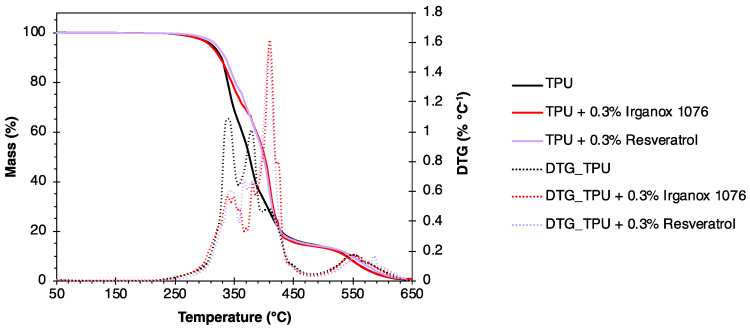
Results of TGA and DTG of TPU-based formulations with 0.3 wt% antioxidants under air.

**Figure 11 polymers-16-00413-f011:**
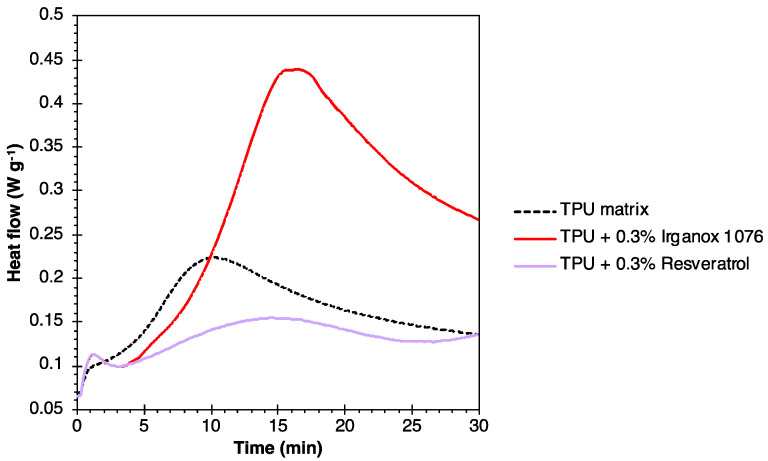
OIT results for TPU-based formulations.

**Figure 12 polymers-16-00413-f012:**
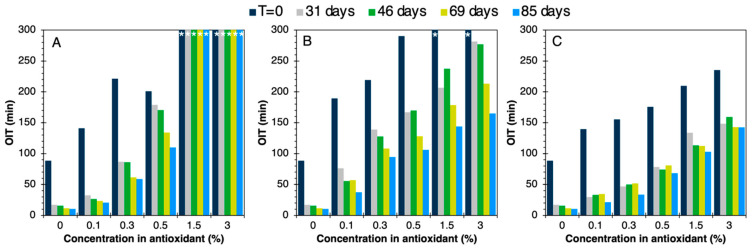
OIT evolutions during aging of TPO formulations with (**A**) Irganox 1076, (**B**) resveratrol, (**C**) BDO-3,4DHB. (*) corresponds to OITs higher than 300 min.

**Figure 13 polymers-16-00413-f013:**
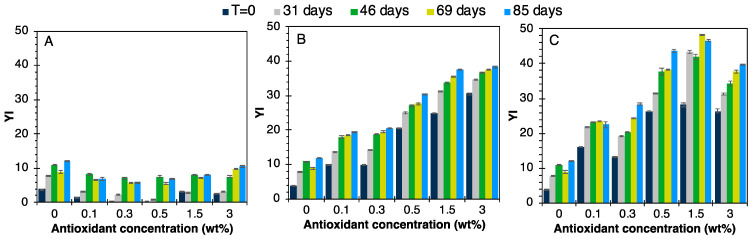
YI evolution during the aging of TPO formulations with (**A**) Irganox 1076, (**B**) resveratrol, (**C**) BDO-3,4DHB.

**Table 1 polymers-16-00413-t001:** Formulations based on TPO and TPU with different antioxidants and concentrations in wt%.

Matrix/Antioxidant	Antioxidant Concentration in wt.%				
TPO + Irganox 1076	0.1	0.3	0.5	1.5	3
TPO + Resveratrol	0.1	0.3	0.5	1.5	3
TPO + BDO-3,4DHB	0.1	0.3	0.5	1.5	3
TPU + Irganox 1076	/	0.3	/	/	/
TPU + Resveratrol	/	0.3	/	/	/

**Table 2 polymers-16-00413-t002:** Properties of neat TPO and the formulations with 3 wt% antioxidant from DSC, TGA and tensile tests.

Sample	T_g_ (°C)	T_m_ (°C)	T_5%_ (°C)	T_deg_ (°C)	σ_max_ (MPa)	E (MPa)	ε_max_ (%)
TPO	−30	143	291	330	17.0 ± 0.3	49.4 ± 2.9	1436 ± 46
TPO + 3% Irganox 1076	−32	149	349	434	21.7 ± 1.0	44.0 ± 3.3	1711 ± 41
TPO + 3% Resveratrol	−32	148	338	432	20.3 ± 1.0	62.7 ± 4.8	1445 ± 30
TPO + 3% BDO-3.4DHB	−31	151	320	435	25.6 ± 1.3	52.2 ± 2.4	1773 ± 40

**Table 3 polymers-16-00413-t003:** T_5%_ and T_deg_ values from TGA of TPU matrix and formulations with 0.3 wt% antioxidants.

Formulations	T_5%_ (°C)	T_deg_ (°C)
TPU	311	340
TPU + 0.3% Irganox 1076	308	409
TPU + 0.3% Resveratrol	319	408

## Data Availability

The data presented in this study are available on request from the corresponding author.

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
