# Peer review of "Synthesis and Assessment of Novel Sustainable Antioxidants with Different Polymer Systems"

_polymers, 2024, doi:10.3390/polym16030413_

Round 1
Reviewer 1 Report
Comments and Suggestions for Authors
This manuscript describes interesting information about phenolic antioxidant, but I think careful correction or check of the following points will be needed.
1. It is difficult to refer to the references. It may be the rule of the publisher, but I think if you want to put the references as number, you should do this according to in order.
2.Figure 1: I think “APTS” should be “PTSA”.
3. Figure 3: I think polymerization occurs by reaction between diol and diisocyanate even if the feed ratio of diol : diicocyanate = 1 : 2. Therefore, I think the figure of Prepolymer should be changed as shown in attached Figure 2-prep.
4. Figure 3: I think chemical structure of written TPU is wrong. I think the figure of Prepolymer should be changed as shown in attached Figure 2-TPU.
5. Figure A4: Where is the signal based on “G”?
6. Figure A5: The integration ratio based on “C”, “D”, and “E” seems to be strange. This should be checked.
7. Lines 324 and 329: I think “20 min” is not correct. This seems to be “25 min.”

Reviewer 2 Report
Comments and Suggestions for Authors
The manuscript concerns with a preparation and performance of new antioxidants from renewable resources. The topic of the study is relevant as new alternatives to fossil-based polymer additives are highly desirable. A whole series of novel antioxidant has been synthesized and their activity has been characterized using a DPPH assay. Based on these measurements one most promising species, BDO-3,4DHB, has been used for stabilization study on PE/PP blend and TPU together with classical antioxidant Irganox 1076 and resveratrol, a natural polyphenol compound.
The manuscript is well structured and written in a clear way. All methods used are described with sufficient details. The overall quality of the paper is very good.
I have only several minor comments and questions to the manuscript.
1. How do you explain rather high increase in Young’s modulus associated with the addition of low amount of Irganox and resveratrol (Fig. 8)? From the DSC curves in Figs 7A and 7B it does not seem that there is a pronounced increase in TPO crystallinity. Together with rather scattered modulus data shown in Fig. A10, I am afraid that the modulus measurements performed at rather high stretching rate of 20 mm.min-1 do not allow to draw any reasonable conclusion from these data.
2. Are you sure that the TPO used does not contain a stabilization system from the producer (it is not obvious from TDS). The presence of other stabilizers could affect the performance of your novel compounds?
3. OIT measurements on TPU should be performed at lower temperatures, because some oxidative processes occur just from the beginning of the measurements. Thus, the exact OIT value is questionable in this case. Nevertheless, the differences between the antioxidants used are clear and the respective statements probably holds.
According to the comments above I recommend the acceptation of the manuscript after a minor revision.
Reviewer 3 Report
Comments and Suggestions for Authors
The paper entitled “Synthesis and assessment of novel sustainable antioxidants with different polymer systems” is devoted to an urgent problem, namely, to the potential of antioxidant systems for polymer materials. In addition, the authors developed a protocol to evaluate variations in polymer properties and estimate the evolution of the potential of different antioxidants over time and with aging. The protocol is of high practical importance. However, the following issues should be clarified and addressed before publication:
1. In line 264, the authors mention a value “a*”, which is used neither in equation (2) nor before. The text and equations should be checked.
2. Captions to axes and legend to Fig. 4 should be clearer.
3. The font of captions to all figures should be unified and corresponded to the font of the main text of the paper.
4. Authors should carefully check the manuscript to remove typos, for example, a period at the end of the title (line 3), etc.
Round 2
Reviewer 1 Report
Comments and Suggestions for Authors
The manuscript has been revised satisfactorily, however, I think some revisions and checks seem to be needed.
1. There are many references, therefore, some errors may be found in references and citations. Please check carefully again.
2. Figure A4: “H” exists without “G”. Is it OK?
3. Figure 5, lines 311 and 316: If “20 min” is not mistake, I think the x-axis of Figure 5 should be revised for “0, 5, 10, 15, 20” for understandability.
Reviewer 2 Report
Comments and Suggestions for Authors
The authors have answered satisfactorily my comments and questions. I recommend publication of the manuscript in the present form.
